# A Multilevel Spatial and Spectral Feature Extraction Network for Marine Oil Spill Monitoring Using Airborne Hyperspectral Image

Jian Wang [1], Zhongwei Li [1], Junfang Yang [1,*], Shanwei Liu [1], Jie Zhang [1,2] and Shibao Li [1]

1   College of Oceanography and Space Informatics, China University of Petroleum (East China), Qingdao 266580, China
2   First Institute of Oceanography, Ministry of Natural Resources, Qingdao 266061, China
*   Correspondence: yangjunfang@upc.edu.cn; Tel.: +1-786-422-9460

**Abstract:** Marine oil spills can cause serious damage to marine ecosystems and biological species, and the pollution is difficult to repair in the short term. Accurate oil type identification and oil thickness quantification are of great significance for marine oil spill emergency response and damage assessment. In recent years, hyperspectral remote sensing technology has become an effective means to monitor marine oil spills. The spectral and spatial features of oil spill images at different levels are different. To accurately identify oil spill types and quantify oil film thickness, and perform better extraction of spectral and spatial features, a multilevel spatial and spectral feature extraction network is proposed in this study. First, the graph convolutional neural network and graph attentional neural network models were used to extract spectral and spatial features in non-Euclidean space, respectively, and then the designed modules based on 2D expansion convolution, depth convolution, and point convolution were applied to extract feature information in Euclidean space; after that, a multilevel feature fusion method was developed to fuse the obtained spatial and spectral features in Euclidean space in a complementary way to obtain multilevel features. Finally, the multilevel features were fused at the feature level to obtain the oil spill information. The experimental results show that compared with CGCNN, SSRN, and A2S2KResNet algorithms, the accuracy of oil type identification and oil film thickness classification of the proposed method in this paper is improved by 12.82%, 0.06%, and 0.08% and 2.23%, 0.69%, and 0.47%, respectively, which proves that the method in this paper can effectively extract oil spill information and identify different oil spill types and different oil film thicknesses.

**Keywords:** hyperspectral remote sensing; oil spill type identification; oil film thickness detection; multilevel spatial and spectral feature; deep learning

## 1. Introduction

Marine oil spills not only endanger marine life and the marine environment but also threaten human health and social and economic development. The types of oil spills on the sea surface are closely related to the formulation of pollution control programs. Oil film thickness is an important parameter for estimating oil spills. Accurately identifying the types of oil spills on the sea surface and quantifying the oil film thickness is of great importance to the emergency treatment of oil spill accidents and the assessment of losses. Therefore, the identification and monitoring technology of offshore oil spills has become important for domestic and foreign scholars.

Hyperspectral remote sensing [1–4] is one of the main means of oil spill monitoring in the ocean. Hyperspectral images (HSI) consist of hundreds of continuous spectral bands and are rich in spectral and spatial information. Early hyperspectral image classification models often utilized traditional machine-learning methods, such as Support Vector Machine (SVM) [5], Multiclass Logistic Regression (MLR) [6], and K-Nearest Neighbor (KNN) [7], and some dimensionality reduction methods based on spectral features, such

as Principal Component Analysis (PCA) [8], Independent Component Analysis (ICA) [9], and Linear Discriminant Analysis (LDA) [10]. However, these methods ignore the connection between neighboring pixels and do not make use of the spatial information of the image, so the classification is not effective. Later on, some joint spectral-spatial classifiers emerged, which can use both spectral and spatial information for classification, such as 3D spectral/spatial Gabor [11], Support Vector Machine based on Markov Random Field (SVM-MRF) [12], Multiclass Multiscale Support Tensor Machine (MCMS-STM) [13], and Multiple Kernel-Based SVM [14]. Although these methods have improved the classification accuracy to a certain extent, the above methods are usually only shallow models with simple extracted features, and the classification results obtained are generally poor.

With the continuous development of deep learning, more and more deep learning models are being used to deal with hyperspectral image classification problems. These models can be broadly divided into network models based on spectral features and joint spectral and spatial feature network models. The first approach based on spectral feature extraction focuses on spectral information, for example, Deep Convolutional Neural Networks (DCNN) [15], Deep Residual Involution Networks (DRIN) [16], Depth-wise Separable Convolution Neural Networks with Residual connection (Des-CNN) [17], and Generative Adversarial Networks (GAN) [18] of the model. Approaches based on the extraction of spectral information ignore the importance of spatial information, such as the extraction of edge information. Secondly, many scholars focus on methods that combine spectral and spatial features, for example, CSSVGAN [19]; SATNet [20]; SSRN [21]; SSUN [22]; DBMA [23]; DBDA [24]; DCRN [25]; MSDN-SA [26]; ENL-FCN [27]; SSDF [28]; and other models.

Many domestic and foreign scholars have carried out research work on hyperspectral oil spill detection [29], oil spill type identification [30,31], and oil film thickness estimation [3,32] using machine learning and in-depth learning methods. Initially, models such as Support Vector Machines (SVM) [31], K-Nearest Neighbor (KNN), and Least Squares (PLS) were widely used for hyperspectral oil spill image classification tasks due to their intuitive oil and water classification results. However, most of them use only hand-crafted features that do not represent the specific distribution characteristics of oil and water. To address this problem, a range of deep learning models such as Convolutional Neural Networks (CNN) [33], Deep Neural Networks (DNN) [34], and Deep Convolutional Neural Networks (DCNN) [35] have been proposed to optimize oil and water classification results by making full use of and abstracting limited data to reduce the number of spectra. Although these methods have made great strides, the learning of oil spill features is not comprehensive enough and there is still much room for improvement in detection accuracy. To fully learn the characteristics of the oil spill, many scholars have developed a deep learning model for oil spill monitoring combined with several methods. For example, Jiang et al. [32] proposed OG-CNN to invert oil film thickness; Wang et al. [36] proposed SSFIN; Jiang et al. [37] proposed an ALTME optimizer; and Yang et al. [38] developed a decision fusion algorithm of deep learning and shallow learning for marine oil spill detection.

In summary, current oil spill monitoring using hyperspectral images mainly extracts spectral and spatial features based on a single level in Euclidean space. Despite the good results, the spectral and spatial information of the oil spill image has not been fully exploited. Some problems do not adequately express the differences between spectral and spatial information. Therefore, it is an important research point to develop joint multilevel spectral and spatial feature extraction methods. For this paper, a multilevel spatial and spectral feature extraction network was developed. The method was applied to the hyperspectral data of outdoor oil spill simulation experiments in UAV, and the validity of the method for oil spill type and oil spill film thickness classification was verified. To evaluate the effectiveness of the proposed algorithm in this study, we compared the effectiveness of the proposed method for oil spill type identification and oil film thickness classification with three mainstream deep learning methods, SSRN, CGCNN, and A2S2KResNet. The

experimental results of the proposed method obtained better classification performance and had higher identification and classification accuracy.

The main contributions of our research are as follows.

- We developed a dimension reduction composition module based on independent component analysis and superpixel segmentation. Pixels with the same spatial and spectral characteristics can be assembled into superpixel blocks to effectively focus irregular oil spill edge information.
- The spectral features were extracted by graph convolution of the spectral domain; the graph attention network assigns weights to different graph nodes to extract the main spatial features, and the features in Euclidean space are extracted using modules based on convolutional neural network architecture. On this basis, the feature fusion algorithm was designed to fuse each part of the extracted features separately to obtain multilevel features, and further fuse the multilevel features at the feature level.

## 2. Proposed Method

In this paper, a novel multilevel spatial and spectral feature extraction network (GCAT) is proposed for marine oil spill type identification and oil film thickness classification. The network framework is divided into three main parts: superpixel graph construction, spectral and spatial information extraction and supplementation, and multilevel feature fusion and classification. Figure 1 illustrates the architecture of the proposed GCAT method for oil spill monitoring. The proposed model structure is based on the Pytorch framework, and the specific structural parameters were set as shown in Table 1.

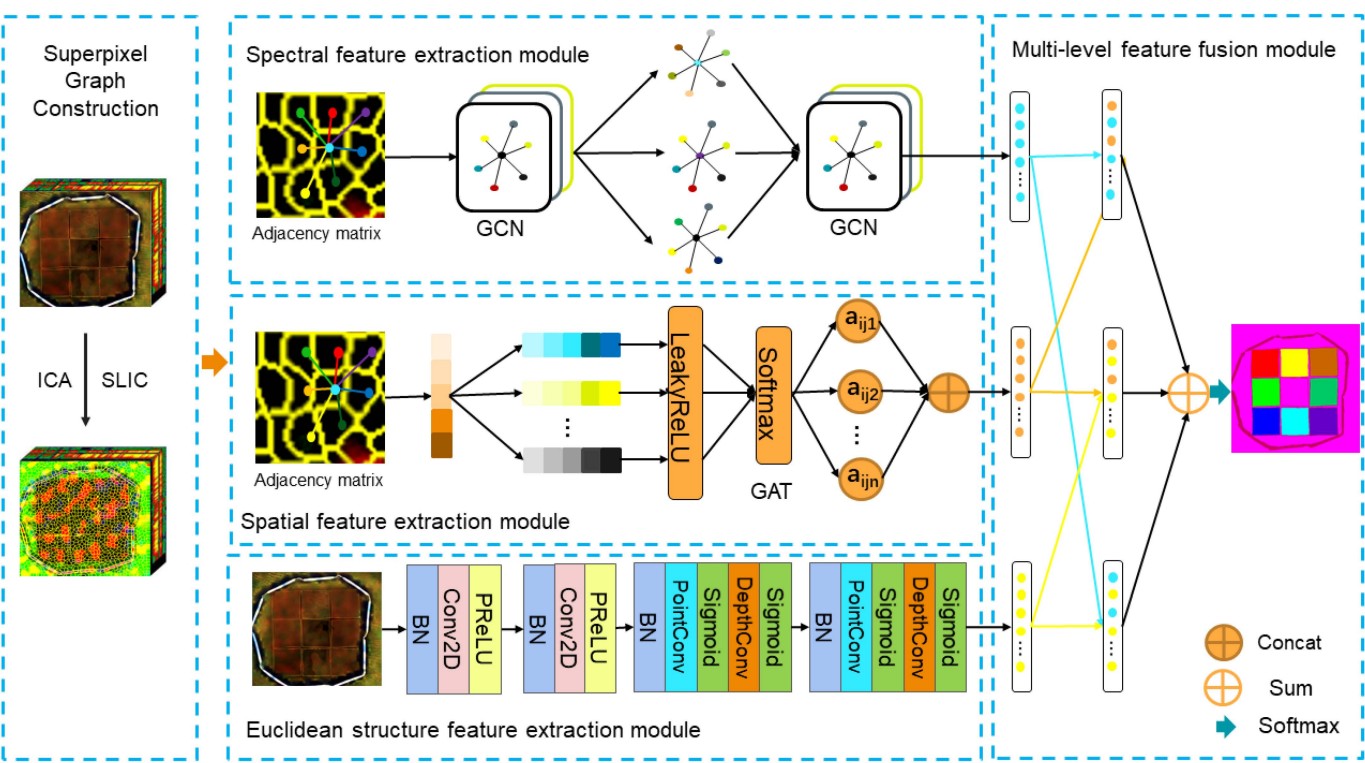

**Figure 1.** The architecture of the proposed model for oil spill monitoring.

**Table 1.** Detailed configuration of the proposed network structure.

| Layer | Input/Output | Kernel | Dilation | Activation |
|---|---|---|---|---|
| GCN layer-1 | (128,128) | - | - | PReLU |
| GCN layer-2 | (128,64) | - | - | PReLU |
| GAT layer-2 | (128,64) | - | - | ELU |
| 2DConv layer-1 | (128,128) | $3 \times 3$ | 2 | PReLU |
| 2DConv layer-2 | (128,128) | $3 \times 3$ | 2 | PReLU |
| DeepConv layer-1 | (64,64) | $5 \times 5$ | - | Sigmod |
| PointConv layer-2 | (128,64) | $1 \times 1$ | - | Sigmod |
| Full connected layer | (64, Class_count) | - | - | - |

Hyperspectral oil spill images have rich spectral and spatial characteristics, and hyperspectral oil spill image data are multidimensional data. In the superpixel graph construction part, we reduce the dimension by ICA. The redundant bands in the oil spill image are removed, and the bands with more information are retained. The segmented superpixel blocks can effectively focus on the edge information of the oil spill image, which is helpful to distinguish different types of oil spills.

In the spectral branch, the graph convolutional neural network converts the spatial graph signal to the spectral domain. The spectral features of the non-Euclidean structure data (graph structure data) after superpixel segmentation are learned in the spectral domain. Its weights are shared, and its parameters are shared. As the number of layers increases, information about distant neighbors accumulates. The more layers, the larger the receptive field, and the more spectral information is involved in the operation, the more fully the spectral information of the oil spill can be learned.

In the spatial branch, the graph attention neural network can perform convolution on the spatial structure of the graph structure data. The attention mechanism is constructed by the features of nodes and neighbor nodes to calculate the edge weight of the central node and neighbor nodes. The masked self-attentional layers were introduced to improve the computational efficiency and process the features of all neighbor pairs in parallel. For each node, the corresponding hidden information is calculated, and the attention mechanism is introduced when calculating its adjacent nodes, improving the model's ability to generalize to unknown graphs. In the process of oil spill spatial feature extraction, the weights are shared and do not depend on the number of nodes in the input graph.

In the Euclidean space structure data, the BCP module (BN-2DConv-PReLU) is designed to enhance the oil spill features. The BPSDS module (BN-PointConv-Sigmoid-DepthConv-Sigmoid) is used to extract the oil spill information. Among them, depth convolution and point convolution consider both channels and regions, which can effectively extract spectral and spatial information.

Traditional oil spill image classification tasks are mostly based on a single method to extract oil spill information and mainly apply CNN to extract features in Euclidean structure. A single method cannot focus on both spatial and spectral information about oil spills. Moreover, the spectral and spatial information differences cannot be fully expressed, and the oil spill information extraction is incomplete. We propose a variety of methods for multilevel feature extraction while focusing on two different forms of structural feature information, and multilevel feature fusion is proposed to combine the extracted features. The oil spill information is fully mined to obtain more detailed oil spill characteristics.

*2.1. Superpixel Graph Construction*

This part mainly performs dimensionality reduction and normalization operations on the data by independent component analysis (ICA) and segmentation of hyperspectral images into superpixel maps by linear iterative clustering (SLIC).

A hyperspectral image contains hundreds of thousands of pixel points, which increases the computational complexity of the subsequent graph neural network and classification. To solve this problem, ICA is first used to perform the dimensionality reduction operation

on the hyperspectral image. Simple Linear Iterative Clustering (SLIC) [39], one of the superpixel segmentation algorithms, continuously iterates to cluster the original N pixels of the image into K superpixel blocks by using a K-mean clustering algorithm, each of which represents an irregular region with strong spectral spatial similarity, and treats each superpixel block as a graph node, thus greatly reducing the number of graph nodes. The features of each node are the average spectral features of the pixels in the superpixels, and the SLIC algorithm effectively preserves the local structure information, which helps the subsequent classification to be accurate. The superpixel segmentation algorithm is defined as follows:

$$
\begin{aligned}
d_{color} &= \sqrt{\left(l_j - l_i\right)^2 + \left(a_j - a_i\right)^2 + \left(b_j - b_i\right)^2}, \\
d_{spatial} &= \sqrt{\left(x_j - x_i\right)^2 + \left(y_j - y_i\right)^2}, \\
D' &= \sqrt{\left(\frac{d_{color}}{N_{cmax}}\right)^2 + \left(\frac{d_{spatial}}{N_{max}}\right)^2}.
\end{aligned}
\tag{1}
$$

Here, $l$, $a$, and $b$ are color values, $d_{color}$ stands for color distance, and $d_{spatial}$ stands for spatial distance. $N_{max}$ is the maximum intra-class spatial distance, defined as $N_{max} = sqrt(N/K)$ and applied to each cluster. $N_{cmax}$ is the maximum color distance.

### 2.2. Spectral and Spatial Feature Extraction

The spectral and spatial information extraction and supplementation section consist of three main branches, which are used to extract spectral and spatial information and Euclidean structure information, respectively, where GCN extracts the spectral features in the non-Euclidean space transformed from the spatial domain to the spectral domain, GAT extracts the spatial features in the non-Euclidean space of the pre-processed image, and CNN is used to capture features in Euclidean space as a complement to the above spatial and spectral features.

#### 2.2.1. Spectral Feature Extraction

A graph is a complex non-linear structure used to describe a one-to-many relationship in non-Euclidean space. Kipf et al. proposed the concept of GCN in 2017. The construction of graph models in GCN [40] relies heavily on the creation of undirected graphs, which are used to describe the set of nodes and edges of a graph structure, as well as the adjacency matrix, which consists of connected nodes with similarity weights between edges.

In this paper, we define the relationship of spectral features in HSI as an undirected graph of $G = (V, E)$, where $V$ denotes the set of nodes $V = \{v_1, v_2, \ldots, v_N\}$, and $E$ is the set of edges. The adjacency matrix defined as $A$ is used to describe the internal associations between nodes. The elements $A_{i,j}$ in $A$ denote the weights of the edges between node $v_i$ and node $v_j$ and are defined as follows:

$$
A_{i,j} = exp\left(-\frac{\|x_i - x_j\|^2}{\sigma^2}\right)
\tag{2}
$$

where $\sigma$ is the parameter controlling the width of the radial basis function, and the vectors $x_i$, $x_j$ represent the corresponding spectral features of the graph nodes $v_i$ and $v_j$ determined by the superpixel segmentation, respectively.

After this, we can solve for the corresponding Laplace matrix $L$ which is shown below.

$$
L = D - A
\tag{3}
$$

where $D$ is the degree matrix of the adjacency matrix $A$.

A more robust representation of graph structure data can be obtained by normalizing the Laplace matrix, which has the real symmetric positive semidefinite property

$$L = I_N - D^{-1/2}AD^{-1/2} \tag{4}$$

Firstly, to perform a nodal embedding of G, a spectral filtering on the graph is defined, which can be expressed as a signal $x$ with a filter in the Fourier domain $g_\theta = diag(\theta)$, the product of

$$g_\theta * x = U_\theta U^\top x \tag{5}$$

where $U$ is the normalized Laplacian matrix $L = I - D^{-(1/2)}AD^{-(1/2)} = U\Lambda U^\top$ of the eigenvector matrix, where $\Lambda$ denotes the matrix of $L$ the diagonal matrix of eigenvalues, and $I$ denotes a unit matrix of suitable size. $g_\theta$ can be understood as an $L$ as a function of the eigenvalues of $g_\theta(\Lambda)$.

To reduce the number of parameters in (4), Kipf [40] et al. used the Chebyshev polynomial $T_k(x)$ to approximate the truncated expansion of $g_\theta(\Lambda)$, up to order $K$.

$$g_{\theta'}(\Lambda) \approx \sum_{k=0}^{K} \theta_k' T_k(\overline{\Lambda}) \tag{6}$$

where $\theta'$ denotes the vector of Chebyshev coefficients, and $\overline{\Lambda} = (2/\Lambda_{max})\Lambda - I$, where $\Lambda_{max}$ is the $L$ the largest eigenvalue of the Chebyshev polynomial which, according to the literature, can be defined as $T_k(x) = 2xT_{k-1}(x) - T_{k-2}(x)$, where $T_0(x) = 1$, and $T_1(x) = x$. We, therefore, define the signal $x$ of the convolution is defined as

$$g_{\theta'} * x \approx \sum_{k=0}^{K} \theta_k' T_k(\tilde{L})x \tag{7}$$

where $\tilde{L} = (2/\Lambda_{max})L - I$ is the scaled Laplace matrix. By limiting $k = 1$, and $\Lambda_{max} \approx 2$ [40], so (7) reduces to

$$g_\theta * x \approx \theta_0'x + \theta_1'(L - I)x = \theta_0'x - \theta_1'D^{-\frac{1}{2}}AD^{-\frac{1}{2}} \tag{8}$$

for which $\theta_0'$ and $\theta_1'$ are two variable parameters, and overfitting can be effectively prevented by reducing the number of parameters, so (8) is further reduced to

$$g_\theta * x \approx \theta\left(I + D^{-\frac{1}{2}}AD^{-\frac{1}{2}}\right)x \tag{9}$$

By order $\theta = \theta_0' = -\theta_1'$, since the $I + D^{-\frac{1}{2}}AD^{-\frac{1}{2}}$ the eigenvalues are within [0,2], repeated use would result in the gradients in the deep network exploding or disappearing. To solve this problem, Kipf and Welling performed a reformulation of $I + D^{-(1/2)}AD^{-(1/2)} \rightarrow \tilde{D}^{-\{1/2\}}\tilde{A}\tilde{D}^{-(1/2)}$, where $\tilde{A} = A + I$ and $\overline{D}_{ii} = \sum_j \overline{A}_{ij}$.

The expression for the final graph convolution is

$$H^{(i)} = \sigma\left(\overline{A}^{(l-1)}W^{(l)}\right) \tag{10}$$

where $H^{(i)}$ denotes the output of the first $l$ the output of the layer, and $\sigma(\cdot)$ denotes the activation function. We denote the extracted spectral features as $H_{spectral}$.

2.2.2. Spatial Feature Extraction

Graph Attention Neural Networks (GAT) [41] operates directly in the spatial domain, with stacked layers that enable nodes to participate in the features of their neighbors, and can assign different weights to different nodes in the neighborhood without any costly matrix operations.

In the preprocessing part, we obtain the superpixel graph, considering each superpixel block in it as a graph node, where the input is a set of node features $V = \{v_1, v_2, \ldots, v_N\}$, where $N$ is the number of nodes. With all nodes sharing the self-attention mechanism, the

attention coefficient between nodes *i* and *j* is calculated, which indicates the importance of node *i*'s features to node *j*. The resulting attention coefficient is

$$e_{ij} = a(Wv_i, Wv_j). \tag{11}$$

This coefficient can represent the importance of node *j* relative to node *i*. To capture the boundary information more accurately, we use a first-order attention mechanism, i.e., only node *j* is connected to node *i*.

Next, we use the softmax function to normalize the attention coefficient to the weight information as

$$\alpha_{ij} = softmax(e_{ij}) = \frac{\exp(e_{ij})}{\sum_{k \in N_i} \exp(e_{ik})}, \tag{12}$$

where $N_i$ denotes the neighborhood of node *i*. Finally, we encapsulate this process into a single-layer feedforward neural network using the LeakyReLU activation function. We then have

$$\alpha_{ij} = \frac{\exp(leaky\,ReLU(a^T[Wv_i\|Wv_j]))}{\sum_{k \in N_i} \exp(leaky\,ReLU(a^T[Wv_i\|Wv_k]))} \tag{13}$$

where $\|$ is the tandem operation. $a^T$ is the transpose of *a*, denoting the learnable parameter. Thus, the node embedding can be expressed as

$$v_i' = \sigma\left(\sum_{j \in N_i} \alpha_{ij}Wv_j\right) \tag{14}$$

To make the node embedding a stable representation of node *i*, we apply the multi-headed attention mechanism at the first attention layer, i.e., execute Equation (14) *K* times independently, and then concatenate the obtained node embeddings to obtain the following output node feature representation as

$$v_i' = \|_{k=1}^K \sigma\left(\sum_{j \in N_i} \alpha_{ij}^k W^k v_j\right) \tag{15}$$

Here, we denote the final acquired features as $H_{spatial}$.

### 2.2.3. Euclidean Structure Feature Extraction

This module is mainly implemented by 2D dilation convolution and depth-separable convolution, consisting of depth convolution and point convolution. The 2D expanded convolution has a larger perceptual field and better feature extraction capability than conventional convolution under the same computational conditions. The depth-separable convolution can effectively reduce the number of parameters required compared to ordinary convolution. Unlike ordinary convolution, which considers both channels and regions, depth-separable convolution achieves the separation of channels and regions.

The features extracted by the 2D expanded convolution and the depth-separable convolution can be expressed as

$$F^{l+1}(X_n) = f\left(W^{l+1} \cdot X_n + b^{l+1}\right) \tag{16}$$

where $W^{l+1}$ is the weight matrix, $b^{l+1}$ is the bias, and $X_n$ is the feature matrix of each layer, and $f(\cdot)$ denotes the activation function. The extracted features are represented as $H_{Euclidean}$.

### 2.3. Multilevel Feature Fusion Module

In the multilevel feature fusion and classification part, the acquired spectral, spatial, and complementary features are fused, and the fused features are fused again to obtain the

final features, which are feedback for the full-connected layer, and the final classification results are output through softmax.

We fuse the extracted spatial features, spectral features, and Euclidean structure features to obtain multilevel features. The calculation formula is as follows.

$$H_1 = H_{spectral} \cdot H_{spatial} \tag{17}$$

$$H_2 = H_{spectral} \cdot H_{Euclidean} \tag{18}$$

$$H_3 = H_{Euclidean} \cdot H_{spatial} \tag{19}$$

where $\cdot$ operation is feature stitching, and finally, the obtained multilevel features are summed to obtain the final $H_{sum} = H_1 + H_2 + H_3$. The classification result is output by the softmax classifier, using the cross-entropy loss function, calculated as follows.

$$L(y, p) = -\frac{1}{N} \sum_{i=1}^{N} \sum_{C=1}^{C} y_{ic} \log(p_{ic}) \tag{20}$$

where $y$ denotes the true value, $p$ is the predicted value for each pixel, and $y_{ic}$ denotes the label $y$ of the $c$ element of the label. $p_{ic}$ denotes that the pixel $i$ belongs to the first $c$ class, calculated by the softmax function, the probability that $C$ and $N$ denote the total number of classes and the total number of samples in the training dataset.

## 3. Experiments and Results

In this paper, two outdoor simulated oil spill scenarios were designed, and an airborne hyperspectral imaging system was applied to obtain typical oil spill type data and oil film thickness data. Based on this, oil type identification and oil film thickness classification experiments were carried out using the proposed method mentioned above and compared with advanced methods such as CGCNN, SSRN, and A2S2KResNet. To evaluate the performance of the proposed model, three evaluation metrics, overall accuracy (OA), average accuracy (AA), and Kappa coefficient, were utilized.

### 3.1. Data

(1)    Oil type data

The oil type detection experiment designed in this paper was carried out in an outdoor seawater pool (40 m × 40 m × 2 m), where the different oil types were separated using PVC panels of 1 m × 1 m. The oil type data were acquired by Cubert-S185 unmanned hyperspectral sensor at an altitude of 15 m (spatial resolution of 6 mm) at noon on 23 September 2020. The image has 125 spectral bands, with a spectral range of 450–950 nm. The preprocessed image size is 500 × 500, with 250,000 labeled samples, including 10 types of crude oil, fuel oil, diesel oil, palm oil, and gasoline. Crude oil was obtained in Shengli Oilfield, China. Fuel oil is the engine fuel of large ships. Crude oil and fuel oil are heavy oils. Diesel is the fuel of choice for high-speed diesel engines on small ships. The gasoline was #95 gasoline, which is similar to the condensate oil that leaked in the East China Sea in 2018. Palm oil is the largest vegetable oil produced in the world in terms of production, consumption, and trade. Diesel, gasoline, and palm oil are all light oils. The hyperspectral false color image and ground truth image of different oil types are shown in Figure 2. The ground truth image of oil type data was produced based on a combination of field photographs and human–computer interactive interpretation.

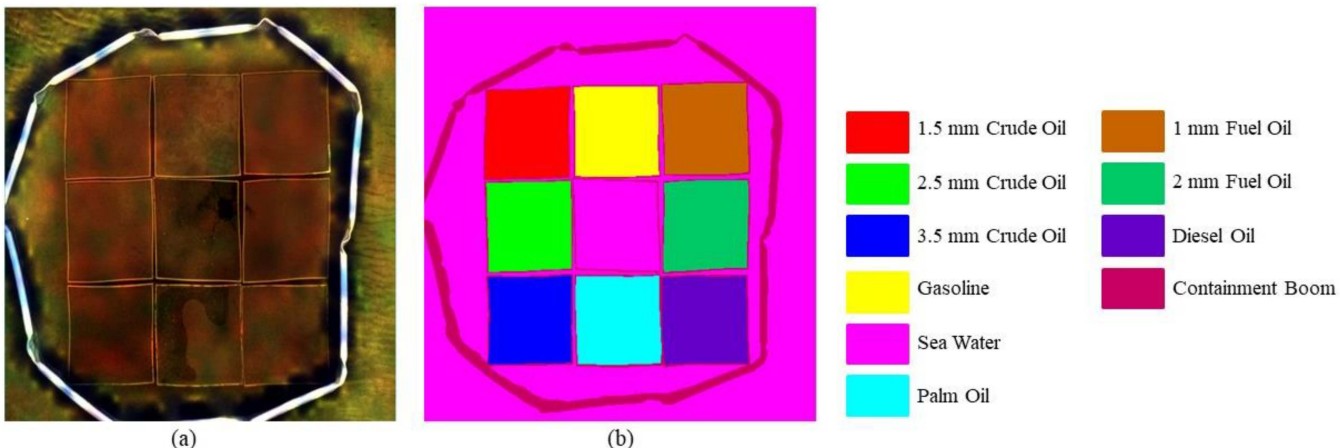

**Figure 2.** Oil type data: (**a**) False-color image (R: 10, G: 23, B: 42); (**b**) ground-truth image.

(2) Oil film thickness data

The oil film thickness detection experiment designed in this paper was carried out in an outdoor black square tank (76 cm × 56 cm × 26 cm) loaded with seawater, with different thicknesses of oil film separated using a PE ring with an inner diameter of 7 cm. The oil film thickness data were acquired by Cubert-S185 airborne hyperspectral sensor at an altitude of 10 m (spatial resolution of 4 mm) at 14:00 on 6 September 2022. The image has 125 spectral bands in the spectral range of 450–950 nm. The preprocessed image size is 96 × 150, with 4778 labeled samples, including seawater and 17 oil films of different thicknesses. The hyperspectral false color image and ground truth image of different thickness oil films are shown in Figure 3. The ground truth image of the oil film thickness data was produced based on a combination of field photographs and human–computer interactive interpretation.

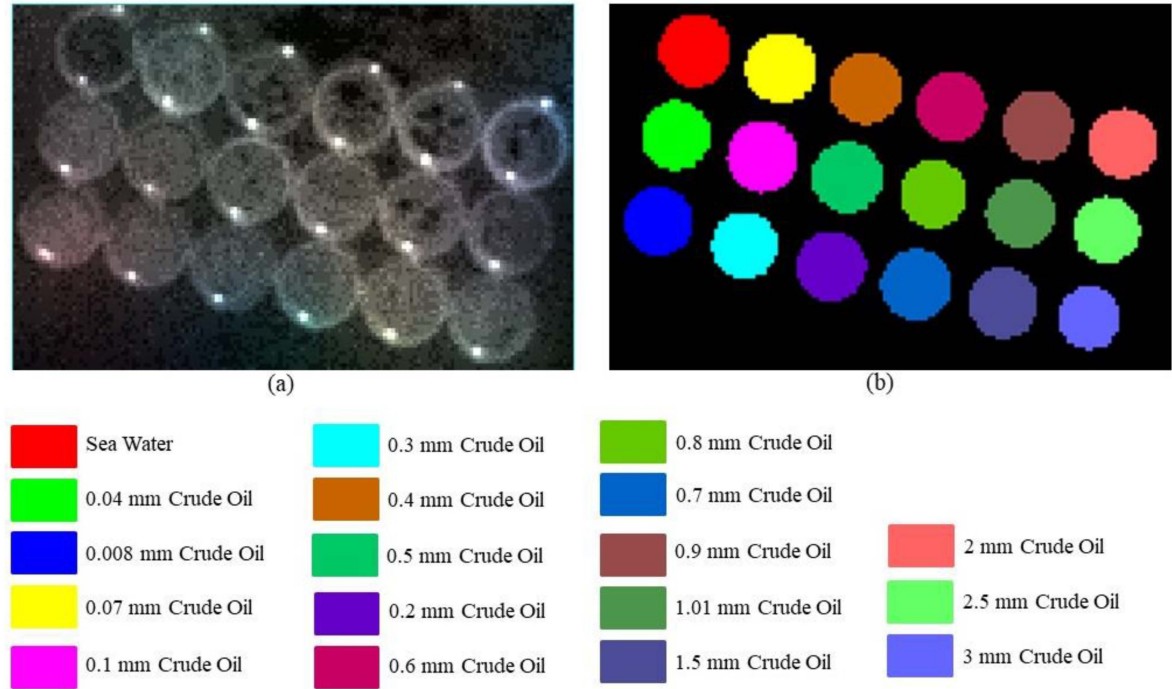

**Figure 3.** Oil film thickness data: (**a**) False-color image (R: 11, G: 27, B: 45); (**b**) ground-truth image.

*3.2. Experimental Setting*

In the outdoor simulated oil spill experiment, we used an anemometer to measure the wind speed of the experimental environment. The experimental data in this paper were collected under the conditions of cloudless and relatively stable wind speed, ensuring that the oil film in the enclosure was relatively evenly distributed to avoid interference caused by the unstable wind speed.

For this paper, we selected several well-known methods of deep learning for comparison, including SSRN [21], CGCNN [42], and A2S2KResNet [43]. To ensure the fairness of the comparison experiments, we used the same hyperparameter settings for these methods, and all experiments were executed on an NVIDIA GeForce RTX 3090 GPU with 24 GB of memory. In this paper, we randomly selected a few samples from each dataset for training and validation. Specifically, for the oil type data, we selected 5% of the samples for training and 5% for validation. For the oil film thickness data, we selected 5% of the samples for training and 5% for validation. Tables 2 and 3 show the number of training, validation, and testing samples for the two types of data.

**Table 2.** Sample numbers of training, validation, and testing in the oil type data.

| Number | Class | Train | Validation | Test | Total |
|--------|-------|-------|------------|------|-------|
| 1 | 1.5 mm Crude Oil | 562 | 562 | 10,117 | 11,241 |
| 2 | 2.5 mm Crude Oil | 572 | 572 | 10,294 | 11,438 |
| 3 | 3.5 mm Crude Oil | 551 | 551 | 9911 | 11,013 |
| 4 | Gasoline | 575 | 575 | 10,352 | 11,502 |
| 5 | Seawater | 6796 | 6796 | 122,328 | 135,920 |
| 6 | Palm Oil | 568 | 568 | 10,231 | 11,367 |
| 7 | 1 mm Fuel Oil | 582 | 582 | 10,473 | 11,637 |
| 8 | 2 mm Fuel Oil | 575 | 575 | 10,345 | 11,495 |
| 9 | Diesel Oil | 564 | 564 | 10,152 | 11,280 |
| 10 | Containment Boom | 1155 | 1155 | 20,797 | 23,107 |
| | Total | 12,500 | 12,500 | 225,000 | 250,000 |

**Table 3.** Sample numbers of training, validation, and testing numbers in the oil film thickness data.

| Number | Class | Train | Validation | Test | Total |
|--------|-------|-------|------------|------|-------|
| 1 | Seawater | 14 | 14 | 259 | 287 |
| 2 | 0.04 mm Crude Oil | 13 | 13 | 237 | 263 |
| 3 | 0.008 mm Crude Oil | 13 | 13 | 235 | 261 |
| 4 | 0.07 mm Crude Oil | 14 | 14 | 250 | 278 |
| 5 | 0.1 mm Crude Oil | 14 | 14 | 249 | 277 |
| 6 | 0.3 mm Crude Oil | 12 | 12 | 219 | 243 |
| 7 | 0.4 mm Crude Oil | 14 | 14 | 255 | 283 |
| 8 | 0.5 mm Crude Oil | 14 | 14 | 253 | 281 |
| 9 | 0.2 mm Crude Oil | 13 | 13 | 242 | 268 |
| 10 | 0.6 mm Crude Oil | 13 | 13 | 242 | 268 |
| 11 | 0.8 mm Crude Oil | 13 | 13 | 224 | 250 |
| 12 | 0.7 mm Crude Oil | 14 | 14 | 250 | 278 |
| 13 | 0.9 mm Crude Oil | 14 | 14 | 251 | 279 |
| 14 | 1.01 mm Crude Oil | 13 | 13 | 236 | 262 |
| 15 | 1.5 mm Crude Oil | 14 | 14 | 246 | 274 |
| 16 | 2 mm Crude Oil | 13 | 13 | 241 | 267 |
| 17 | 2.5 mm Crude Oil | 12 | 12 | 219 | 243 |
| 18 | 3 mm Crude Oil | 11 | 11 | 194 | 216 |
| | Total | 238 | 238 | 4302 | 4778 |

*3.3. Experimental Results*

The oil spill simulation experiment in this paper was carried out under relatively controlled outdoor conditions. The oil was spread over a long time to ensure that the oil

film thickness was relatively uniform. The collected UAV hyperspectral data were obtained under relatively stable illumination and wind speeds. These considerations ensured that data acquisition was not affected by other factors.

The proposed method and the other three algorithms were applied to the data obtained from the outdoor simulated oil spill scenario. The data were hyperspectral data of oil spill type and hyperspectral data of oil film with different thicknesses. The oil type recognition results (Figure 4) and the oil film thickness classification results (Figure 5) were obtained. From the oil type identification results (Figure 4), CGCNN showed a poor ability to distinguish the same crude oil with different thicknesses, and it was easy to classify them into the same class, such as classifying 2.5 mm crude oil into 1.5 mm crude oil; and the classification effect for diesel oil was also poor; SSRN and A2S2KResNet showed misclassification in each class boundary, such as classifying some oil boundaries as seawater. From the oil film thickness classification results (Figure 5), CGCNN also had difficulty in distinguishing some oil films thicknesses, such as dividing 0.4 mm oil film into 0.07 mm, 0.5 mm, and 0.6 mm thicknesses; SSRN and A2S2KResNet had the phenomenon of dividing thinner oil films into thicker oil films, such as dividing 0.07 mm oil film into 0.4 mm oil film, and 0.1 mm oil film into 0.8 mm oil film. GCAT was obvious in each class boundary, and the misclassification phenomenon was reduced, thanks to the superpixel composition based on ICA and SLIC and the multilevel fusion features extracted by GCAT, which can better focus on the spectral information and retain the class boundary information.

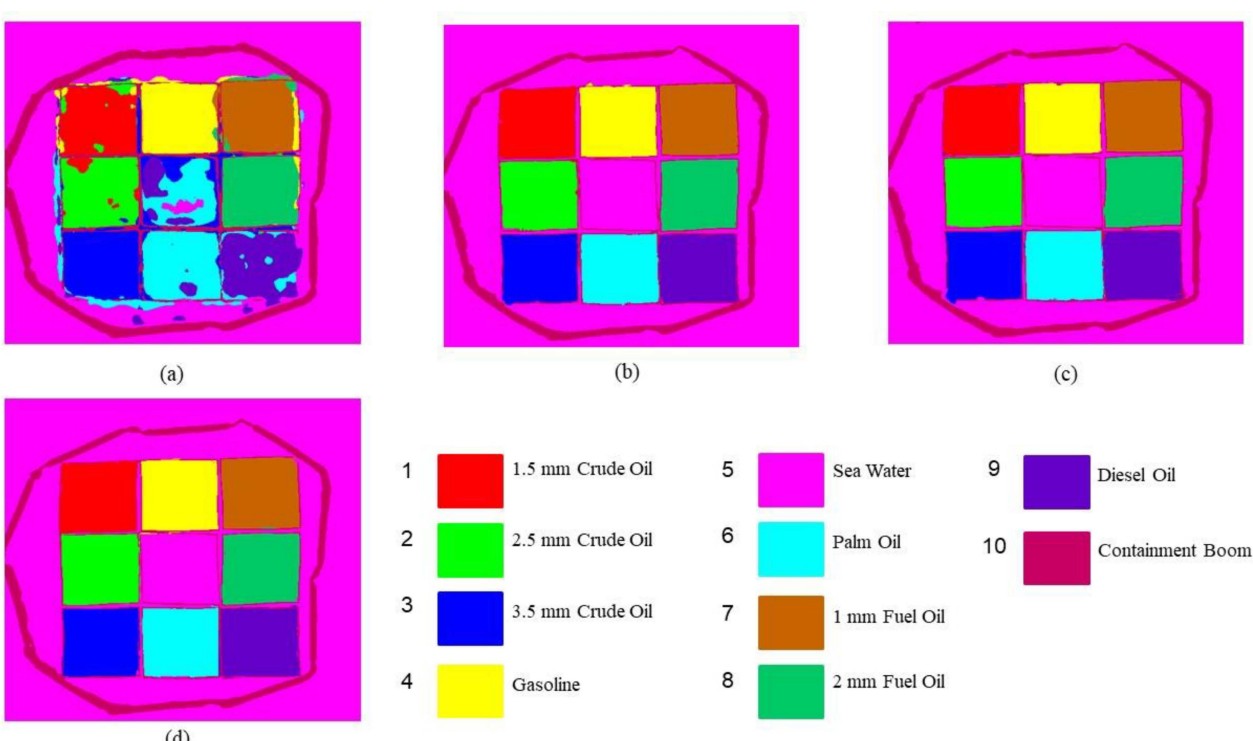

**Figure 4.** Identification results of the oil type data. (**a**) CGCNN, (**b**) SSRN, (**c**) A2S2KResNet, (**d**) GCAT.

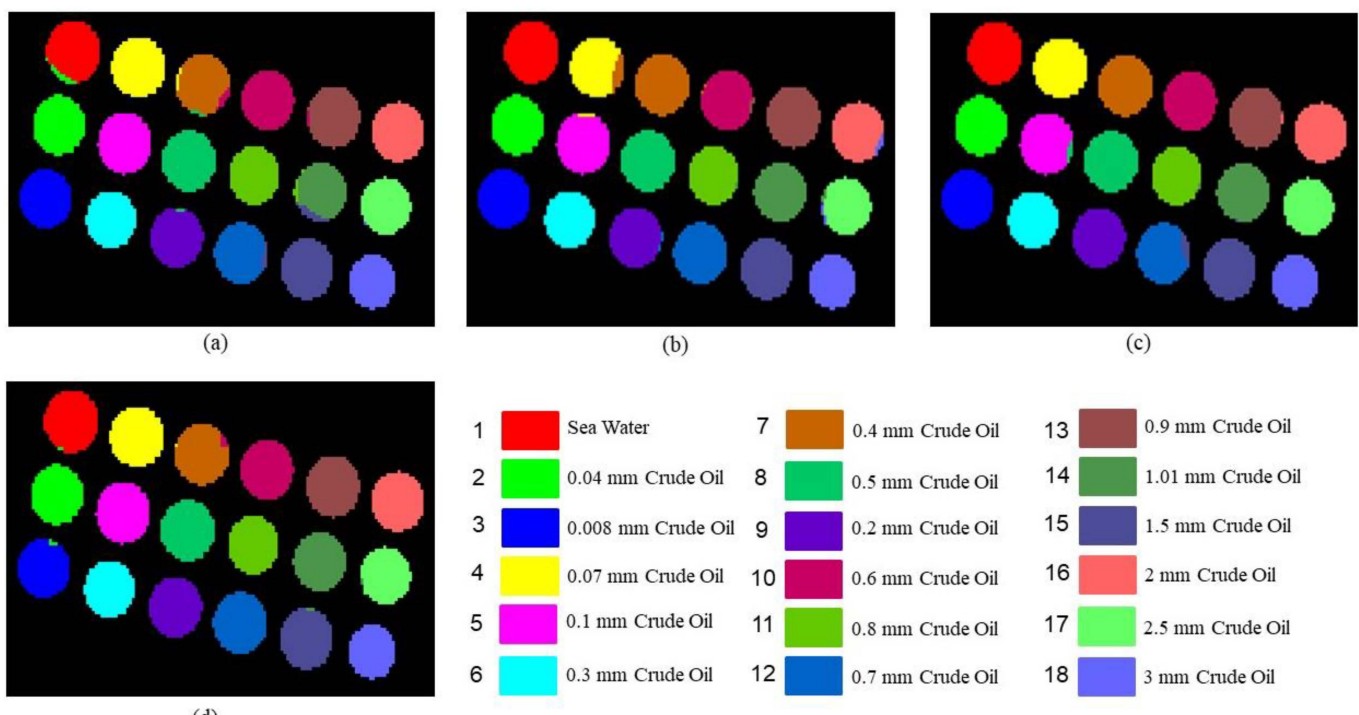

**Figure 5.** Detection results of the oil film thickness data. (**a**) CGCNN, (**b**) SSRN, (**c**) A2S2KResNet, (**d**) GCAT.

Evidently, the proposed GCAT method achieved the best performance in oil type identification and oil film thickness classification compared with the other three methods, which proves the effectiveness of our method in oil spill type identification and oil film thickness classification.

To evaluate the performance of the proposed model, the overall oil type identification accuracy and film thickness classification accuracy were evaluated using overall accuracy (OA), average accuracy (AA), and Kappa coefficient. Additionally, the recall was used to evaluate the classification accuracy of a single oil type or a single film thickness. The oil type recognition accuracy and film thickness classification accuracy are shown in Tables 4 and 5.

**Table 4.** Classification results of different methods for the oil type data.

| Number | Class | CGCNN | SSRN | A2S2KResNet | GCAT |
|--------|-------|-------|------|-------------|------|
| 1 | 1.5 mm Crude Oil | 90.77 | 99.62 | 99.95 | 99.13 |
| 2 | 2.5 mm Crude Oil | 82.98 | 99.94 | 99.83 | 99.53 |
| 3 | 3.5 mm Crude Oil | 94.82 | 99.41 | 99.08 | 99.05 |
| 4 | Gasoline | 97.85 | 99.14 | 99.98 | 99.31 |
| 5 | Seawater | 81.85 | 99.22 | 99.26 | 99.31 |
| 6 | Palm Oil | 92.62 | 98.92 | 98.96 | 97.97 |
| 7 | 1 mm Fuel Oil | 92.12 | 99.86 | 99.87 | 99.16 |
| 8 | 2 mm Fuel Oil | 96.87 | 99.66 | 99.21 | 98.25 |
| 9 | Diesel Oil | 84.44 | 99.06 | 98.70 | 98.88 |
| 10 | Containment Boom | 87.32 | 93.18 | 92.81 | 94.88 |
| | OA (%) | 85.98 | 98.74 | 98.72 | **98.80** |
| | AA (%) | 90.16 | **98.80** | 98.76 | 98.65 |
| | Kappa Coefficient | 0.8052 | 0.9814 | 0.9811 | **0.9824** |

Note: The bolded values represent the optimal values.

**Table 5.** Classification results of different methods for the oil film thickness data.

| Number | Class | CGCNN | SSRN | A2S2KResNet | GCAT |
|--------|-------|-------|------|-------------|------|
| 1 | Seawater | 95.33 | 100.00 | 100.00 | 99.69 |
| 2 | 0.04 mm Crude Oil | 100.00 | 100.00 | 100.00 | 100.00 |
| 3 | 0.008 mm Crude Oil | 96.89 | 100.00 | 100.00 | 99.66 |
| 4 | 0.07 mm Crude Oil | 100.00 | 98.66 | 100.00 | 100.00 |
| 5 | 0.1 mm Crude Oil | 99.62 | 100.00 | 99.46 | 99.35 |
| 6 | 0.3 mm Crude Oil | 98.33 | 100.00 | 100.00 | 100.00 |
| 7 | 0.4 mm Crude Oil | 94.29 | 94.64 | 95.72 | 98.51 |
| 8 | 0.5 mm Crude Oil | 100.00 | 100.00 | 97.71 | 99.92 |
| 9 | 0.2 mm Crude Oil | 97.69 | 99.79 | 100.00 | 100.00 |
| 10 | 0.6 mm Crude Oil | 94.46 | 100.00 | 100.00 | 99.34 |
| 11 | 0.8 mm Crude Oil | 100.00 | 100.00 | 100.00 | 99.91 |
| 12 | 0.7 mm Crude Oil | 96.95 | 99.04 | 100.00 | 99.61 |
| 13 | 0.9 mm Crude Oil | 93.41 | 98.81 | 99.74 | 99.45 |
| 14 | 1.01 mm Crude Oil | 88.60 | 96.08 | 98.37 | 99.65 |
| 15 | 1.5 mm Crude Oil | 100.00 | 98.63 | 95.71 | 99.76 |
| 16 | 2 mm Crude Oil | 100.00 | 100.00 | 100.00 | 98.67 |
| 17 | 2.5 mm Crude Oil | 97.61 | 98.64 | 100.00 | 99.44 |
| 18 | 3 mm Crude Oil | 99.90 | 96.45 | 99.16 | 99.58 |
| | OA (%) | 97.35 | 98.89 | 99.11 | **99.58** |
| | AA (%) | 97.39 | 98.93 | 99.22 | **99.59** |
| | Kappa Coefficient | 0.9720 | 0.9882 | 0.9906 | **0.9956** |

Note: The bolded values represent the optimal values.

Firstly, from the classification results of oil type data, the algorithm that only uses CNN to extract features (CGCNN) achieved general classification accuracy, especially for the classification results with seawater, probably because the spatial and spectral features were not extracted separately. Additionally, using only one way to extract features at the same time may cause part of the spatial or spectral feature information to be ignored, making the final results poor; SSRN and A2S2KResNet can recognize most of the categories, but their classification results are not good for containment booms with relatively few pixels; our proposed GCAT still maintains a high recall rate in the category with few pixel points. Secondly, from the classification results of oil film thickness data, CGCNN showed poor classification results in some categories, such as for 1.01 mm crude oil. The algorithms with joint spectral spatial feature extraction (SSRN, A2S2KResNet) showed better performance, which indicates that it is desirable to extract spectral and spatial information separately to achieve classification. Finally, from the classification results, it can be seen that the GCAT model combining spatial and spectral features extracted in non-Euclidean space with those extracted in Euclidean space was effective, achieving the best OA, Kappa, and competitive AA in both datasets.

Compared with a single method, the multilevel spatial spectral feature extraction network proposed by us can obtain more complete oil spill information through multilevel feature fusion. The results of each type of oil and each type of oil film thickness were more continuous, and there were no more fragmented patches (results of misclassification). This is because we fused the proposed features twice. First, we carried out a multilevel feature fusion, fused each part of the acquired feature information in pairs to obtain more detailed features, and then spliced the fused features to obtain the final output oil spill feature. This output feature focuses on the important spectral and spatial information in the oil spill image. In addition, the spilled oil and oil film boundaries in our method results are obvious, which is due to the independent component analysis (ICA) dimensionality reduction and superpixel segmentation method. The oil spill image was divided into superpixel blocks with high spectral spatial similarity through superpixel segmentation. The boundary of each superpixel block is obvious, and the edge information was fully learned.

Among these compared methods, GCAT improved the overall recognition accuracy in oil type data by at least 12.82%, 0.06%, and 0.08%, and the overall classification accuracy

for oil film thickness data by at least 2.23%, 0.69%, and 0.47%. The classification results show that by simultaneously using the spatial and spectral features in non-Euclidean space, and the features in Euclidean space, the information in the oil spill images can be fully explored to improve the classification accuracy. Additionally, the accuracy of each category is improved without the phenomenon that one category is very high or very low in the comparison methods, and the algorithm is highly stable.

## 4. Discussion

### 4.1. Influence of the Proportion of Training Samples on Classification Results

In this section, several experiments are designed to explore the robustness of the proposed method under different training ratios. Two datasets were randomly selected with 1%, 3%, and 5% training samples. Figures 6 and 7 show the results of the four methods on the two datasets with different training ratios.

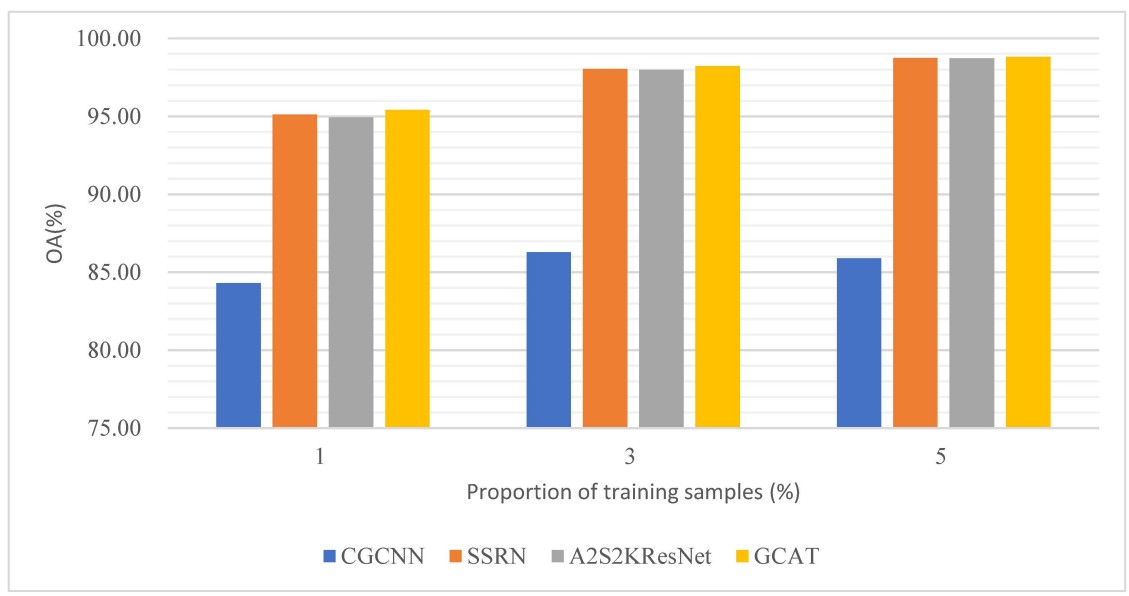

**Figure 6.** Classification results on the oil type data with different proportions of training samples.

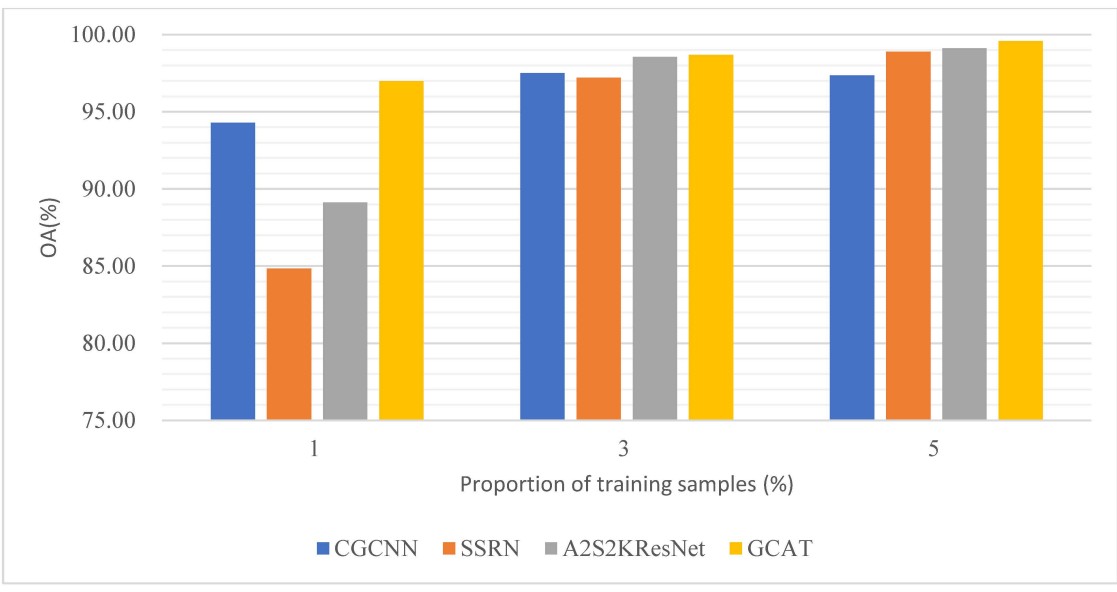

**Figure 7.** Classification results on the oil film thickness data with different proportions of training samples.

Firstly, it is clear that different proportions of training samples resulted in different classification performances for the four methods: as the proportion of training samples increased, the classification accuracy increased, and the proposed GCAT method achieved the best performance compared to the other methods with different samples.

Secondly, in the oil film dataset, GCAT still showed better oil film classification performance than CGCNN, SSRN, and A2S2KResNet for a small sample size of 1%.

Finally, it can be seen from Figures 6 and 7 that as the training ratio increases, the accuracy of the combined spatial and spectral methods (SSRN and A2S2KResNet) became closer and closer, especially for oil data. Compared with these, our proposed GCAT is still highly competitive.

### 4.2. Comparison with the Single-Level Feature Fusion Module

Taking the oil type data as an example, to further verify the effectiveness of the proposed multilevel feature fusion module, the recognition accuracy of the single-level feature fusion module was evaluated separately, and the results are shown in Table 6 (the best results are bolded).

**Table 6.** The OA (%), AA (%), and Kappa coefficient for ablation experiments.

| Modules | OA (%) | AA (%) | Kappa |
|---|---|---|---|
| Y1 | 91.26 | 91.87 | 0.8712 |
| Y2 | 90.65 | 91.59 | 0.86 |
| Y3 | 98 | 97.89 | 0.9705 |
| Y1·Y2 | 91.29 | 91.96 | 0.8717 |
| Y1·Y3 | 97.23 | 96.95 | 0.9593 |
| Y2·Y3 | 97.98 | 98.05 | 0.9703 |
| Y1·Y2+ Y1·Y3+ Y2·Y3 | **98.80** | **98.65** | **0.9824** |

Where Y1 is the spectral feature extraction module, Y2 is the spatial feature extraction module, and Y3 is the Euclidean spatial feature extraction module.

The results show that spectral features and spatial features were expressed differently, and the effect of extracting only a certain part of features alone was not optimal. GCAT improved OA by 7.54%, 8.15%, and 0.8%, respectively, in comparison with a single-feature extraction module, and 7.51%, 1.57%, and 0.82% in comparison with a two-by-two fusion-feature extraction module. It is further demonstrated that multilevel feature fusion helps oil spill information extraction, which can significantly improve classification accuracy and has better robustness.

### 4.3. Comparison with Other Works

This paper also focuses on oil type identification and oil film thickness classification. The difference from other studies is mainly reflected in two aspects: (1) In terms of field experiment setup, for oil type identification experiments, the experimental scene was carried out in a real seawater pool. The selection of oil products was also based on past oil spill events. For the oil film thickness experiment, the design of the oil film thickness was a more detailed division of the oil film thickness mentioned in the Bonn agreement. The data were obtained under relatively stable conditions of light and wind speed. (2) In terms of the model, we used a variety of methods to extract multilevel features from the oil spill image. The extracted spatial and spectral features were fused at multiple levels. Compared with the features extracted by a single method, the fused features can better express the oil spill information. From the results of model experiments, our method achieved good accuracy in oil spill identification and oil film thickness classification.

## 5. Conclusions

Marine oil spill accidents occur frequently and cause great harm to marine ecology. The effective identification of oil spill type and the accurate quantification of oil film thickness are the prerequisites for the emergency response and damage assessment of oil spill accidents. In this paper, by designing an outfield oil spill simulation experiment, we acquired hyperspectral images of five typical oil spill types and 17 different oil film thicknesses based on an unmanned airborne hyperspectral imaging system and proposed a marine oil spill monitoring model with multilevel feature extraction suitable for oil spill scenarios. In this paper, we mainly draw the following two conclusions: (1) To address the problem of incomplete extraction of oil spill information by single-level features, we designed the graph convolutional network module to focus on spectral information, the graph attention network module to focus on the main spatial information, and the module based on convolutional neural network architecture to focus on the oil spill information in the Euclidean space. A multilevel feature fusion method was developed to obtain multilevel features by fusing the obtained spatial and spectral features and the features of Euclidean space. Compared with the single-level feature extraction method, the proposed method shows better oil type identification accuracy and oil film thickness classification accuracy. (2) For the spectral and spatial feature differences in different oil spill images, a method of fusing multilevel features at the feature level is proposed, which can more fully express the spectral and spatial differences in oil spill type and oil film thickness images. By comparing with the mainstream methods such as CGCNN, SSRN, and A2S2KResNet, the overall classification accuracy of the proposed method was improved by 12.82%, 0.06%, and 0.08%, and 2.23%, 0.69%, and 0.47%, respectively, and the classification results also achieve the best visual effect, which proves the effectiveness and robustness of the proposed method.

Our model was designed to solve the problem of incomplete extraction of oil spill information at a single level. The hyperspectral image of the oil spill was classified and identified by combining the spectral and spatial characteristics of non-Euclidean space with the information on the oil spill characteristics of Euclidean space. Although the final output of the model was better than other classical neural networks, there were some problems encountered in the actual classification. For example, the accuracy of the GCAT model proposed in this paper was less prominent when there are few types of ground objects in the oil spill hyperspectral image. The problem may be that the composition of the data preprocessing section was not detailed enough to classify fewer and scattered categories into other categories. Therefore, constructing a more effective feature map is the next step to improve the accuracy of oil spill identification and oil film thickness classification.

In this study, multilevel feature extraction and fusion were carried out for the hyperspectral images of oil spills acquired by unmanned airborne hyperspectral sensors. We plan to carry out future experiments on oil spill detection based on unmanned airborne multi-sensors to acquire multisource remote sensing data such as hyperspectral, SAR, and radar. The acquired multisource remote sensing data will be used to further validate the multilevel feature extraction and fusion method and improve the marine oil spill detection capability.

**Author Contributions:** Conceptualization, J.W. and Z.L.; methodology, J.W., Z.L. and J.Y.; software, J.W., S.L. (Shanwei Liu) and J.Y.; validation, J.W. and J.Y.; formal analysis, J.W.; data curation, J.W.; writing—original draft preparation, J.W.; writing—review and editing, J.W., Z.L., S.L. (Shanwei Liu) and S.L. (Shibao Li); supervision, Z.L. and S.L. (Shibao Li); project administration, Z.L., J.Z., S.L. (Shanwei Liu) and J.Y.; funding acquisition, J.Z., Z.L. and J.Y. All authors have read and agreed to the published version of the manuscript.

**Funding:** This research was funded by the National Natural Science Foundation of China (Grant No. U1906217, No. 61890964, No. 42206177), Shandong Provincial Natural Science Foundation (Grant No. ZR2022QD075), Qingdao Postdoctoral Application Research Project (Grant No. qdyy20210082), and the Fundamental Research Funds for the Central Universities (Grant No. 21CX06057A).

**Data Availability Statement:** The data used in this study are available on request from the first author.

**Acknowledgments:** The authors thank the reviewers and editors for their positive and constructive comments, which have significantly improved the work.

**Conflicts of Interest:** The authors declare no conflict of interest. The funders had no role in the design of the study; in the collection, analyses, or interpretation of data; in the writing of the manuscript, or in the decision to publish the results.

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
