# Peer review of "A Multilevel Spatial and Spectral Feature Extraction Network for Marine Oil Spill Monitoring Using Airborne Hyperspectral Image"

_remotesensing, doi:10.3390/rs15051302_

Round 1

Reviewer 1 Report

In this paper, the authors proposed a new deep learning model to address the problem of marine oil spill monitoring. They provided a thorough comparison of several existing state-of-the-art approaches, showing that their proposed network outperforms all of them. Overall, the paper is well-written, and the results are promising. The authors have provided a clearly-defined problem and shown that their proposed network is effective in solving it. I suggest the authors revise the manuscript based on the following points prior to considering it for publication.

Line 36: HSI: please use the full name in the first appearance of the acronyms.

Why doesn’t this research consider oil spill lookalikes? I don’t see any analysis in this regard! How may the lookalikes affect the results of your methodology?

Introduction: it seems that some citations do not match the finding of the literature, and some are only done based on keywords. Moreover, I suggest you use the more recently published oil spill detection review papers to provide a comprehensive introduction.

Section 2.2.1: I think the content of this section is misleading as it gives the impression that the GCN concept is being introduced for the first time in this paper. It may be that the authors are presenting a new application of GCN or a novel variation of the model, but the GCN concept itself is not new, and it has been widely studied and used in previous research. In academic writing, it is important to be clear and accurate in stating the contribution and novelty of the research, to avoid misleading readers and to accurately credit previous work in the field.

Section 3.3.2 (Experimental Setting): Did you consider environmental effects such as wind speed in your analysis?

Section 3.3 (Experimental Results): The paper could be further strengthened by a more detailed analysis of the results, especially in terms of understanding why the proposed network achieves such promising results.

Line 311-315: A long and difficult-to-follow sentence! Please consider revising.

Discussion: The discussion should be broadened to give the implication of the work compared to other works.

Conclusion: The contribution of the developed approach should be discussed in-depth with a brief statement on the limitations encountered and knowledge gaps, and suggest ideas for future directions.

Author Response

Dear reviewer,

Reviewer 2 Report

Marine oil spills can cause serious damage to marine ecosystems and biological species, and the pollution is difficult to repair in the short term. Accurate oil type identification and oil thickness quantification are of great significance for marine oil spill emergency response and damage assessment. This study proposed a multi-level spatial and spectral feature extraction network to accurately identify oil spill types and quantify oil film thickness, and perform better extraction of spectral features and spatial features. Overall, the work is complete, but there are several points to be improved as below:

1. According to the content of the article, the title of the paper should be “A multi-level spatial and spectral feature extraction...” rather than “A multi-level spatial and spatial feature extraction...”. The author should carefully examine the full text to avoid similar mistakes.

2. The research background and significance of the introduction in the paper are too simple to fully highlight the research focus. One good paragraph contains one main idea, where the first sentence summarizes the main idea, with the following sentences elaborate the details. The introduction is suggested to reorganize.

3. In the introduction, the author points out three contributions of the proposed algorithm. The contribution of the algorithm should be the particularity or highlight of the algorithm that is different from other algorithms. The third contribution introduced by the author is the comparison between the proposed algorithm and other algorithms, and should not be the contribution of the proposed algorithm.

4. In this paper, a multi-level spatial and spectral feature extraction network is proposed for the identification of marine oil spills and the classification of oil film thickness. From the author's description of the algorithm, it is not clear where the unique advantages of the proposed algorithm are for oil type identification and oil film thickness classification. Please add this part.

5. In the “3.1 Data” section, the manuscript directly gives the ground truth images of different oil types and different thickness of oil film. The reliability of the ground truth image directly affects the accuracy of the algorithm. The reliability of the two ground truth images is not described in the article. Please supplement it.

6. The designed network model is relatively complex, and whether it contains relatively mature modules applied to image processing or other related fields. Please explain the efficiency of the network in statistical pixel level feature differences when solving the problem of oil spill hyperspectral image classification.

7. It is suggested to add the principle formula of the construction part of the super-pixel image for the readers to understand.

8. There are some format errors in the paper, such as “3.3.2 Experimental Setting” should be “3.2 Experimental Setting”.

Author Response

Dear reviewer,

Reviewer 3 Report

The authors do not explain the metods well enough to judge the soundness of the results.  The slicks, especially in the 1m x 1m squates and even the smaller squares are patch and that is to be expected. That pathiness can be very important to responders and is not reflected in the discussion nor the analyses. There is no expalnation of the specific type of crude (e.g., heavy, light, VLSFO) nor the level of solar radiation/cloud cover etc.  It is difficult to know the validity/accuracy and precision of these methods without knowing these factors.  There should be some independent method to validate that the thicknesses on within the squares are those desired.

Author Response

Dear reviewer,

Round 2

Reviewer 1 Report

The authors have addressed all of my comments, and  I would recommend it be published in the present form.